# The Association of the Exposure to Work-Related Traumatic Events and Work Limitations among Firefighters: A Cross-Sectional Study

**DOI:** 10.3390/ijerph16050756

**Published:** 2019-03-02

**Authors:** Mi-Ji Kim, Yeoju Jeong, Yeun-Soon Choi, Ae-Rim Seo, Yeongmi Ha, Minjeong Seo, Ki-Soo Park

**Affiliations:** 1Department of Preventive Medicine and Institute of Health Sciences, College of Medicine, Gyeongsang National University, Jinju 52727, Korea; mijikim@gnu.ac.kr (M.-J.K.); yeju2002@daum.net (Y.J.); choeys0808@gmail.com (Y.-S.C.); sarim2101@naver.com (A.-R.S.); 2College of Nursing and Gerontological Health Research Center in Institute of Health Sciences, Gyeongsang National University, Jinju 52727, Korea; yha@gnu.ac.kr (Y.H.); mjseo@gnu.ac.kr (M.S.)

**Keywords:** firefighters, psychological trauma, mental health, work performance, work limitations

## Abstract

Many firefighters suffering from traumatic events while on duty participate in the work force. However, the impact of work-related traumatic events on work performance is unclear. To address this issue, we administered a work-related traumatic event experience survey, the health-related work limitations questionnaire, the Korea Depression Scale, and the World Health Organization quality of life assessment to firefighters from four cities in Korea. The health-related work limitations questionnaire measured limitations in their ability to perform three specific work demands—physical, psychosocial, and environmental. Of 685 firefighters enrolled in the study, 618 (90.2%) were included in the final analyses. Their mean age ± standard deviation (SD) was 41.1 ± 8.0 years and mean employment period ± SD was 163.0 ± 101.2 months. A large percentage of participants reported they had experienced work-related traumatic events. Among the firefighters, the total work limitation prevalence was 21.5%, 16.8% of them were limited in their abilities to perform physical work demands, 15.5% were limited in performing psychosocial work demands, and 13.8% were limited in their abilities to function without difficulty within the ambient work environment. Based on multivariable logistic regression, with having work-related traumatic event experience of a threat or injury to self, the odds ratio (OR) of having work limitations in physical (OR = 3.32, 95% CI = 1.17–9.41), psychosocial (OR = 3.05, 95% CI = 1.08–8.61), environmental (OR = 4.89, 95% CI = 1.66–14.40) work demands, and total work limitation (OR = 3.73, 95% CI = 1.44–9.68) increased significantly. With experiences of treating fatalities or injured patients, the OR of total work limitation increased significantly (OR = 2.07, 95% CI = 1.09–3.91). We demonstrated a relationship between the firefighters’ exposure to work-related traumatic events and their work limitations. A professional care management system for firefighters to prevent and manage work-related traumatic events should be developed in order to protect and improve their performance ability.

## 1. Introduction

Work disability is an important public health and social policy issue, because work roles have a central part in the lives of most adults in industrialized nations and function as the engine of the national economy.

In firefighters, it is necessary to manage work limitations not only in terms of personal health and quality of life, but also in terms of the stable management of firefighting organizations and ultimately in providing effective firefighting services to the public.

Unlike other occupations, the work of firefighters involves many physical and mental tasks, which include a lot of risk factors that must be resolved directly. For example, they may have to enter a hot fire while wearing heavy clothing and protective gear, face casualties at a terrible accident scene, or they or their colleagues may suffer a direct physical or mental impairment in a fire or emergency. These traumatic events are known to be particularly serious when considering the incidence of post-traumatic stress disorder (PTSD) among firefighters [1,2]. Exposure to traumatic events is the first criterion for the development of PTSD. In Korea, more than 80% of firefighters are exposed to work-related traumatic events [3]. Post-traumatic stress disorder is one of the mental disorders belonging to anxiety disorder, which has the main symptoms of reaction, arousal, and avoidance related to an incident that occurs after experiencing extreme stress [4]. The prevalence of PTSD in firefighters is 10%–50%, which is higher than the lifetime prevalence of PTSD in the general population, of 7% [5,6].

In general, PTSD is accompanied by depression, anxiety, and alcohol-use disorders; it causes not only physical and mental disorders, but also social and occupational problems [7,8,9,10,11,12,13,14]. It is characterized by cognitive dysfunction, and it may lead to problems of observation and control of information, which interferes with work performance and leads to the limitations of work demands [15,16,17,18]. In addition, it causes symptoms such as fatigue, loss of concentration, loss of interest in activities, sleep disorders, social atrophy, vigilance, and behavioral avoidance, increasing absenteeism and causing problems in efficiency and social relations in the workplace [19].

Although studies on PTSD have been conducted so far, studies on whether the trauma experience itself affects social life (e.g., working) are rare. Most studies on the relationship between traumatic events and work limitations have focused on emergency room personnel and drivers in public transportation [20,21,22,23], and studies on firefighters have been rare. Most of the studies concerning firefighters have investigated whether post-traumatic stress was related to health-related quality of life or to social functions as part of their social roles [5,24,25,26].

The purpose of this study was to investigate the relationship between the exposure to work-related traumatic events during the job duties of firefighters and their work limitations. We studied the extent of the work limitations of firefighters and determined whether exposure to work-related traumatic events was significantly related to work limitations even after adjusting for physical, mental, and social factors.

## 2. Materials and Methods 

### 2.1. Participants and Procedure

The participants of this study were firefighters working in four cities in Gyeongsangnam-do province (Gimhae, Sacheon, Yangsan, and Jinju) in Korea. All firefighters working in the four fire stations were included in the study. The total number of participants was 685, but 22 people were unable to participate because of long-term business trips or sick leave during the survey period, and a further 36 people who had less than a year of employment period were judged by the researchers as less likely to have experienced exposure to job-related injury in the previous year. A total of 627 self-administered questionnaires were distributed. After excluding nine participants, who had missing data in the questionnaire, a total of 618 responses were included in the final analysis (Figure 1).

Data collection of this study was conducted between 2 May 2016 and 24 June 2016 with the approval of the Institutional Review Board (IRB) of Gyeongsang National University (GIRB-A16-Y-0016). We first explained the purpose and contents of the study to the firefighters. The survey was conducted after receiving written informed consent from the participants to voluntarily participate in the research. The time required to complete the questionnaire was 15–20 min. The questionnaires were self-administered and the completed questionnaires were sealed in envelopes to protect personal information.

### 2.2. Measures

#### 2.2.1. Demographic and Occupational Characteristics

The demographic characteristics were sex (‘male’ or ‘female’), age, marital status (‘married’ or ‘single or others’), and education level (‘less than high school graduate’ or ‘college/university graduate or above’). Occupational characteristics were employment period and type of task (‘firefighting’, ‘rescue’, ‘emergency medical service’, or ‘administration or other’).

#### 2.2.2. Work-Related Traumatic Events

The experience of exposure to work-related traumatic events was investigated by the National Emergency Management Agency in Korea [5], which examined 16 trauma cases that were developed through consultation with firefighting public officials. The purpose of their investigation of the 16 cases was to choose one of the most remembered traumatic events that participants experienced on the job. If participants had no most remembered traumatic events, they were classified as ‘none of the traumatic events’. The selection of traumatic events were classified into three types: (1) Threat or injury of self (life-threatening injuries, fear-of-life threats or injuries, exposure to toxic or infectious threats, physical assault, threats or attacks by weapons, intimidation, etc.); (2) death or injury to fellow firefighters; and (3) treating fatalities/injured patients (witnessing and taking care of a terribly injured body, or reconstruction of a severely damaged child). 

#### 2.2.3. Health-Related Work Limitations

Previous studies have focused on physical work limitations when studying social functions, such as firefighters’ exposure to high temperatures [24], sleep disorders [25], and physical training [26]. A health-related quality of life assessment instrument (the short-form 36 health survey questionnaire, SF-36) was used to assess the work function in firefighters when dealing with mental disorders such as PTSD [5,27,28,29]. However, firefighters should be aware of multidimensional task limitations such as physical, psychosocial, and environmental work limitations. Therefore, we developed the health-related work limitations based on the 15-item health-related work limitations questionnaire used by Lerner, Amick III, Malspeis, and Rogers [30]. They measured work limitations at the demand level for several reasons. Specially, a demand-level questionnaire identified the breadth of limitations that may occur as a result of health problems. For example, a large number of individuals have comorbid conditions which may produce a range of diverse physical, psychosocial, and environmental consequences [30].

A questionnaire was developed to assess the content validity of the health-related work limitations measurement tool by one professor of preventive medicine and two professors of nursing in the related field, and to select the item that showed a content validity index (CVI) of 80.0%.

The questionnaire was based on the stem question: ‘During the past four weeks, how much difficulty have you had doing the following work activities because of any ongoing health problems or health concerns?’ The response options were based on a five-point Likert scale: ‘None’, ‘slight’, ‘some’, ‘a great deal’, and ‘not done, cannot do it’. The questionnaire’s 15 items clustered into three distinct groups. One group was ‘physical work demands’, which consisted of six items that associated mainly with physical health and addressed work demands requiring use of the body (e.g., ‘keeping up with required standards of personal appearance, dress, personal safety and hygiene’ or ‘walking or moving around your usual work area or building’). Another group was ‘psychosocial work demands’, which contained five items, addressing a mentally and socially demanding work tasks (e.g., ‘keeping up with requirements for training, certification, licensure, education, or experience’ or ‘maintaining required attitudes toward your work’). A final group of four items on ‘environmental work demands’ was related to adapting to the work environment (e.g., ‘doing required commuting or local and long-distance traveling’) and acquiring and processing sensory information (e.g., ‘doing things that require you to recognize sounds, signals, temperatures, textures, consistencies, or smells’). Separate scale scores were computed for each respondent. A respondent was defined as ‘not limited’ on a scale (score = 0) if either ‘none’ or ‘slight’ was reported on all items contained within that scale. Respondents reporting ‘some’, ‘a great deal’, or ‘not done, cannot do it’ on one or more items contained within the scale were defined as ‘limited’ (score = 1). A respondent was considered as having work limitations if there were limitations in one or more subscales of physical, psychosocial, and environmental work demands.

The overall internal reliability (Cronbach’s α coefficient) of the developed questionnaire was 0.965, and the Cronbach’s α coefficients of each subscale were 0.908 (physical work demands), 0.916 (psychosocial work demands), and 0.877 (environmental work demands), respectively. 

#### 2.2.4. Depression

Depression was measured by the Korea Depression Scale (KDS-30) developed by Lee and Rhee [31]. The Korea Depression Scale is a self-reported depression test that is standardized for Korean subjects, based on depression scales developed abroad. This scale comprises 30 questions, with five questions for each of the following six domains: Negative thoughts about the future, negative thoughts about oneself, anxiety, restlessness, depressive feelings, physical symptoms, and loss of motivation. Each question is rated on a five-point Likert scale, with ‘not at all = 0’ and ‘very much so = 4’. The total score ranges from 0 to 120; and the higher the score, the higher is the level of depression. Cronbach’s α coefficient was 0.957, indicating high internal consistency.

#### 2.2.5. Health-Related Quality of Life

The Korean version of the World Health Organization quality of life assessment instrument (WHOQOL-BREF) was used. Min, Kim, Lee, Jung, Suh, and Kim [32] developed the Korean version of this scale. It consists of 26 items, with seven items in the physical health domain, six items in the psychological domain, three items in the social relationship domain, and eight items in the environment domain. Each item has a five-point Likert scale, and the conversion score per sub-domain is 0–100 points. The higher the score, the better the quality of life related to health.

Cronbach’s α coefficients for each sub-domain of the questionnaire items showed high internal consistency within the physical health domain (0.846), psychological domain (0.876), social relationship domain (0.880), and environmental domain (0.944).

#### 2.2.6. Family Concerns

A questionnaire consisting of four items was used to measure family concerns about the performance of the fire service (anxiety about injuries, concern about trauma case, concern about excessive workload, or economic burden). Each item had a five-point Likert scale (‘not at all’ = 1 and ‘very much so’ = 5) and the mean score was used. The higher the score, the greater is the family concerns. The internal reliability using Cronbach’s α coefficient was found to be 0.822.

### 2.3. Statistical Analyses 

The demographic and occupational characteristics of the participants, health-related work limitations, depression, health-related quality of life, and family concerns by type of traumatic events were analyzed using frequency analysis and descriptive statistics. The work limitations prevalence was expressed as a percentage of ‘limited’ by performing a *C*hi-square (*χ*^2^) test for each subscale. One-way analysis of variance (ANOVA) was used to compare overall mean scores for health-related quality of life and family concerns by type of traumatic events, the Kruskal–Wallis test was used for depression because it was not normally distributed, and post hoc comparisons were made using Tukey’s method.

The risk of work limitations (physical, psychosocial, environmental, and total) according to the type of traumatic events (reference value: None of the traumatic events) was determined using multivariable logistic regression analysis. Model 1 was unadjusted and Model 2 was adjusted for sex, age, marital status, education level, employment period, and type of task. Model 3 was adjusted for sex, age, marital status, education level, employment period, type of task, depression, health-related quality of life, and family concerns.

All analyses were performed with IBM SPSS 24.0 (International Business Machines Corp., Armonk, NY, USA) and a two-tailed *p* < 0.05 was considered significant.

## 3. Results

### 3.1. Demographic and Occupational Characteristics and Type of Traumatic Events of Participants

The demographic and occupational characteristics of participants were analyzed; 97.6% of them were men with a mean age of 41.0 ± 8.0 years (range: 25–59 years). The mean employment period as a firefighter was 163.0 ± 101.2 months (range: 12–444 months). The most frequent type of work was firefighting (44.3%), followed by administrative or others (26.2%), emergency medical service (23.6%), and rescue (5.8%).

Of the 618 respondents, 70.2% replied that they had memories of traumatic events during their work. Of the total respondents, 55.0% said they had memories of treating fatalities/injured patients, 8.4% had threat/injury to self, and 6.8% had death/injury to fellow firefighters (Table 1).

### 3.2. Work Limitation Rates of Participants

The limitations of physical and psychosocial work demands were 16.8% and 15.5%, respectively, and the limitation of environmental work demands was relatively low (13.8%). A number of the participants (21.5%) answered that they had difficulty in performing tasks in at least one area (Table 2).

The limitation of physical work demands was the highest in the group of ‘death/injury to fellow firefighters’ (28.6%), with a significant difference (*p* = 0.035). The limitation of psychosocial work demands was the highest in the ‘threat/injury of self’ group (25.0%) and was statistically significant (*p* = 0.049). The limitation of environmental work demands was also highest in the ‘threat/injury of self’ group (26.9%), which was statistically significant (*p* = 0.004). The overall work limitations was high in the ‘threat/injury of self’ group (32.7%), which was statistically significant (*p* = 0.041) and low in the ‘treating fatalities/injured patients’ group (22.4%), but it was not statistically significant (*p* = 0.578).

### 3.3. Depression, Health-Related Quality of Life, and Family Concerns

Depression was not statistically related to the type of traumatic events. Physical health-related quality of life was the lowest in participants with ‘threat/injury to self’ (64.2 ± 19.5 *p* < 0.001). Psychological health–related quality of life was also lowest in the ‘threat/injury to self’ group (62.3 ± 23.2, *p* < 0.001). Social health–related quality of life was the lowest, with 51.4 ± 29.8, in the case of ‘threat/injury to self’ and was significantly highest in the ‘none of the traumatic events’ group (67.3 ± 22.6, *p* < 0.001), respectively. Quality of life related to environmental health was also the lowest, with a mean of 52.1 ± 24.3, in the case of ‘threat/injury to self’ (*p* < 0.001) The mean of perceived family concerns according to type of traumatic events was the highest, with a mean of 2.9 ± 1.1 in the group of ‘death/injury to fellow firefighters’ (*p* = 0.001) (Table 3).

### 3.4. Relationship between Work Limitations and Work-Related Traumatic Events

Table 4 shows the risk of work limitations according to the type of traumatic events while on duty, based on the group with ‘none of the traumatic events’.

In Model 3, which adjusted for health-related quality of life, depression, and family concerns, there was a statistically significant risk of physical work limitations in the group with ‘threat/injury of self’ (OR = 3.32, 95% CI = 1.17–9.41). In the case of psychosocial work limitations, the ‘threat/injury of self’ group was significant in Model 3 (OR = 3.05, 95% CI = 1.08–8.61). Environmental work limitations were also significant in the ‘threat/injury of self’ (OR = 4.89, 95% CI = 1.66–14.40). In the total work limitations, the statistical significance of the groups ‘threat/injury of self’ and ‘treating fatalities/injured patients’ were significant (OR = 3.73, 95% CI = 1.44–9.68; OR = 2.07, 95% CI = 1.09–3.91, respectively) (Table 4).

## 4. Discussion

Preventing and managing firefighters’ work limitations is important, not only for firefighters but also for public safety. As a result of the study, it was revealed that the type of traumatic events, especially most remembered traumatic events, are related to work limitations in firefighters. That is, the present results demonstrate that the more severe the traumatic event a firefighter has experienced, greater would be the level of work-related stress the firefighter will have.

As a firefighter, the prevalence of experiencing most remembered traumatic events was about 70%, and the most traumatic stress event was remembered as treating fatalities or injured patients. This is similar to the results of previous studies in Korea [4]. Firefighters are exposed repeatedly and over a long-term to trauma because of their jobs. Firefighters, who are exposed to traumatic events, may be prejudiced that they will be immune to the cumulative stress of trauma. However, this actually causes PTSD, and this can lead to emotional exhaustion and work limitations. 

From the results of this study, the prevalence of total health-related work limitations of firefighters was 21.5%. According to their job demands, the limitation of physical work demands was the highest, at 16.8%, followed by psychosocial work demands (15.5%), and environmental work demands (13.8%). In a study of firefighters who had experienced the 911 terrorist attacks in the United States [29], a single-item survey of functional impairment at work showed that about 9% of firefighters reported a decrease in occupational function. The prevalence of work limitations in this study was somewhat higher, but differences in assessment tools make it difficult to draw direct comparisons. Maybe the prevalence was higher in this study since responding anything other than ‘not limited’ in one or more of the 15 questions resulted in a ‘limited’ score. Also, a lack of mental support services for firefighters in Korea can cause work limitations of firefighters. However, in a study using the same questionnaire, 32.4% of workers with chronic conditions in the U.S. were limited in their abilities to perform at least one work demand, and approximately 12–14% of workers who reported no chronic conditions had work limitations [30]. It can be assumed that the work-related trauma experience of firefighters alone may cause work limitations. In addition to exposure to traumatic events, this study also included depression [27,29,33,34,35,36], health-related quality of life [33], and family concerns [33,36] in order to adjust for bias in relation to the traumatic events and work limitations of firefighters. The results showed that firefighters who had experienced dangers and injuries themselves or witnessed terrible injuries (or injured patients) had difficulty in performing their duties. Especially, the ‘threat/injury of self’ group was significant across all areas of work demands. This means that, even after adjusting for physical, mental, and social health, exposure to work-related traumatic events had a significantly negative effect on duty. To reduce the work limitations of firefighters, not only quality of life related to physical, mental, and social health but also exposure of traumatic events should be systematically managed. Various forms of social support (e.g., counseling service) have been found to be protective against traumatic stress. Particularly, the ‘threat/injury to self’ group is likely to have multidimensional work limitations, so immediate and appropriate intervention is needed.

This study has some limitations. First, it is limited to firefighters who are currently working. Therefore, this study’s findings may underestimate the true number of work limitations because the study does not include those who quit work as a result of serious work limitations. Second, the questionnaire in this study evaluates the validity of the work limitation—whether it is a work limitation because of the traumatic events. Although the survey was limited to trauma cases, other physical or mental conditions such as underlying diseases or occupational illness can also cause work limitations and the exact meaning can be diluted during the survey process. Third, the self-reported health-related work limitations questionnaire was the only instrument to assess firefighters’ work limitations in this study. It obscures the validity of the work limitations. Additional instruments such as checking firefighters’ absenteeism or presenteeism, conducting in-depth interviews, or using other work limitations questionnaires, which evaluate the work limitations more objectively are needed. Fourth, as this is a cross-sectional study, it is not easy to deduce causality. That is, work limitations may increase traumatic experiences such as a threat or injury to self. Additional follow-up studies are needed. Finally, in this study, it was impossible to exclude the possibility of recall bias, because of the use of self-report type items for trauma exposure and various other factors. In addition to the severity of traumatic events, frequency of exposure may also be a factor that affects PTSD or occupational functioning in firefighters.

This is the first time, however, that the relationship between the type of traumatic events experienced by firefighters and the resulting multidimensional work limitations has been studied. In the future, longitudinal research on work limitation results of trauma or post-traumatic stress are needed for firefighters. Based on these future studies, professional care management systems to protect and improve firefighters’ performance ability should be developed [37].

## 5. Conclusions

We demonstrated a relationship between the firefighters’ exposure to work-related traumatic events and their work limitations. The exposure to work-related traumatic events of firefighters is connected not only to mental health and post-traumatic stress but also to physical, psychosocial, and environmental work limitations. A professional care management system for firefighters to prevent and manage work-related traumatic events should be developed in order to protect and improve their performance ability.

## Figures and Tables

**Figure 1 ijerph-16-00756-f001:**
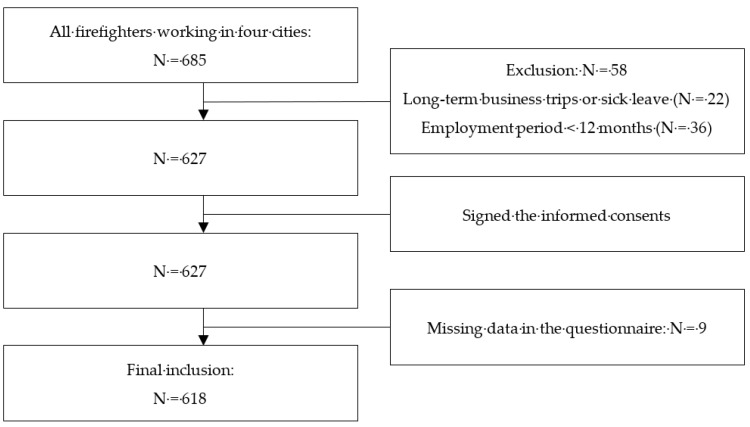
Flowchart of study participation.

**Table 1 ijerph-16-00756-t001:** Demographic and occupational characteristics of participants.

Characteristics	N (%)
*Demographic characteristics*	
Sex	
Male	603 (97.6)
Female	15 (2.4)
Mean age (year) ± SD (range: 25–59)	41.1 ± 8.0
Marital status	
Married	537 (86.9)
Singe or others	81 (13.1)
Education level	
≤ High school	170 (27.5)
≥ College or university	448 (72.5)
*Occupational characteristics*	
Mean employment period (months) ± SD (range: 12–444)	163.0 ± 101.2
Type of task	
Firefighting	274 (44.3)
Rescue	36 (5.8)
Emergency medical service	146 (23.6)
Administration or others	162 (26.2)
Type of traumatic events	
Threat/injury to self	52 (8.4)
Death/injury to fellow firefighters	42 (6.8)
Treating fatalities/injured patients	340 (55.0)
None of the traumatic events	184 (29.8)

Abbreviation: SD, standard deviation.

**Table 2 ijerph-16-00756-t002:** Percentage of firefighters with some or more work limitations caused by work-related traumatic events.

Type of Traumatic Events	Work Limitations							
	Physical		Psychosocial		Environmental		Total	
	N (%)	*p*	N (%)	*p*	N (%)	*p*	N (%)	*p*
Threat/injury of self	13 (25.0)	0.100	13 (25.0)	0.049	14 (26.9)	0.004	17 (32.7)	0.041
Death/injury to fellow firefighters	12 (28.6)	0.035	8 (19.0)	0.515	7 (16.7)	0.570	12 (28.6)	0.249
Treating fatalities/injured patients	58 (17.1)	0.866	55 (16.2)	0.626	45 (13.2)	0.679	76 (22.4)	0.578
None of the traumatic events	21 (11.4)	0.019	20 (10.9)	0.037	19 (10.3)	0.107	28 (15.2)	0.013
Total participants	104 (16.8)		96 (15.5)		85 (13.8)		133 (21.5)	

Statistical significance was tested by the Chi-square (*χ*^2^) test.

**Table 3 ijerph-16-00756-t003:** Depression, health-related quality of life, and family concerns by work-related traumatic events.

Type of Traumatic Events	Depression	Health-Related Quality of Life				Family Concerns
		Physical	Psychological	Social	Environmental	
Threat/injury to self	12.2 ± 13.8	64.2 ± 19.5 ^a^	62.3 ± 23.4 ^a^	51.0 ± 29.9 ^a^	52.1 ± 24.3 ^a^	2.6 ± 1.1 ^a,b^
Death/injury to fellow firefighters	14.5 ± 16.8	65.9 ± 20.0 ^a^	67.2 ± 18.8 ^a,b^	57.7 ± 22.7 ^a,b^	57.5 ± 21.5 ^a,b^	2.9 ± 1.1^a^
Treating fatalities/injured patients	12.6 ± 14.7	68.4 ± 17.7 ^a^	66.7 ± 19.6 ^a^	57.5 ± 22.0 ^a^	58.0 ± 20.6 ^a^	2.4 ± 0.9 ^b^
None of the traumatic events	10.3 ± 13.3	74.2 ± 17.6 ^b^	73.4 ± 18.8 ^b^	67.3 ± 22.6 ^b^	66.0 ± 22.2 ^b^	2.3 ± 0.9 ^b^
Total participants	12.0 ± 14.4	69.6 ± 18.2	68.4 ± 19.9	59.9 ± 23.5	59.9 ± 21.8	2.4 ± 0.9
*p*	0.462	<0.001	<0.001	<0.001	<0.001	0.001

The values are expressed as mean ± standard deviation. Statistical significance was tested by one-way analysis of variance (ANOVA). ^a,b^ The Tukey’s method was applied for post hoc comparisons and different letters indicate significant difference.

**Table 4 ijerph-16-00756-t004:** Multivariable logistic regression odds ratios (OR): Relationship between work limitations and work-related traumatic events.

Types of Traumatic Events	Type of Work Limitations			
	Physical	Psychosocial	Environmental	Total
	OR (95% CI)	OR (95% CI)	OR (95% CI)	OR (95% CI)
Model 1				
Threat/injury of self	2.59 (1.19–5.62)	2.73 (1.25–5.97)	3.20 (1.47–6.95)	2.71 (1.345.48)
Death/injury to fellow firefighters	3.11 (1.38–6.97)	1.93 (0.78–4.74)	1.74 (0.68–4.45)	2.23 (1.02–4.87)
Treating fatalities/injured patients	1.60 (0.94–2.73)	1.58 (0.92–2.73)	1.32 (0.75–2.34)	1.60 (0.99–2.58)
Model 2				
Threat/injury of self	3.16 (1.40–7.13)	3.18 (1.41–7.17)	3.90 (1.72–8.84)	3.17 (1.52–6.63)
Death/injury to fellow firefighters	3.35 (1.43–7.86)	2.03 (0.80–5.19)	1.85 (0.69–4.92)	2.31 (1.02–5.22)
Treating fatalities/injured patients	2.02 (1.14–3.56)	1.96 (1.10–3.48)	1.63 (0.89–2.98)	1.95 (1.18–3.22)
Model 3				
Threat/injury of self	3.32 (1.17–9.41)	3.05 (1.08–8.61)	4.89 (1.66–14.40)	3.73 (1.44–9.68)
Death/injury to fellow firefighters	2.51 (0.86–7.27)	1.13 (0.36–3.54)	1.15 (0.33–3.94)	1.52 (0.54–4.30)
Treating fatalities/injured patients	2.00 (0.99–4.04)	1.72 (0.86–3.44)	1.62 (0.75–3.52)	2.07 (1.09–3.91)

Abbreviations: OR, odds ratio; CI, confidence interval. Model 1 was unadjusted; Model 2 controlled for sex, age, marital status, education level, employment period, and type of task; Model 3 controlled for sex, age, marital status, education level, employment period, type of task, depression, health-related quality of life, and family concerns. Types of traumatic events (reference value: None of the traumatic events); work limitations (not limited = 0, limited = 1), sex (female = 0, male = 1), age (continuous), marital status (married = 0, single or others = 1), education level (college/university graduate or above = 0, high school graduate = 1), employment period (continuous), type of task (reference value: Administration or others), depression (continuous), health-related quality of life (continuous), and family concerns (continuous).

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
