# Peer review of "The Association of the Exposure to Work-Related Traumatic Events and Work Limitations among Firefighters: A Cross-Sectional Study"

_ijerph, 2019, doi:10.3390/ijerph16050756_

Reviewer 1 Report

The authors explored the effect of the exposure to work-related potentially traumatic events on work limitations among firefighters, and found that exposure to traumatic events was significantly associated with work limitations. Although this is an interesting study, there are some points to consider.

The authors should provide detailed information on sampling method. Did firefighters participate voluntarily?

In data analysis section, The authors stated that "the average comparison of... used the ANOVA" (page4, ling170-171). However, this sentence should be changed because ANOVA is not average comparison method. Why the authors used hierarchical logistic regression model? Please, provide further information. In Table 3, It seems that the authors simply used multivariate logistic regression analysis.

Figure 1 is somewhat confusing and the authors should provide real number. I think that table is better than figure.

In table 2, It seems that depression scale might not have a normal distribution. The authors should check normality of variables, and consider nonparametric test. 

Because this study simply explored the association between traumatic event and work limitation using cross-sectional study, correction for multiple comparison (physical, psychosocial, and environmental limitation) might not be necessary in Table 3. However, the author should comment this issue in data analysis section. 

English editing is required for this manuscript. 

Author Response

Response to Reviewer 1 Comments

Point 1: The authors should provide detailed information on sampling method. Did firefighters participate voluntarily?

Response 1: We agree with the reviewer and have added that all subjects participated voluntarily in the research to the participants and procedure section of the manuscript (lines 93-95).

Point 2: In data analysis section, The authors stated that "the average comparison of... used the ANOVA" (page4, ling170-171). However, this sentence should be changed because ANOVA is not average comparison method.

Response 2: We agree with the reviewer and have added the following sentences in the statistical analyses section (lines 183-187): “One-way analysis of variance (ANOVA) were used to compare overall mean scores for health-related quality of life and family concerns by type of traumatic events, Kruskal-Wallis test was used for depression because it was not normally distributed, and post hoc comparisons were made using the Tukey’s method.”

Point 3: Why the authors used hierarchical logistic regression model? Please, provide further information. In Table 3, It seems that the authors simply used multivariate logistic regression analysis.

Response 3: We agree with the reviewer and have changed the expression “hierarchical logistic regression” to “multivariable logistic regression” in the statistical analyses section (line 190) and the Table 3 (Table 4 in the revised version).

Point 4: Figure 1 is somewhat confusing and the authors should provide real number. I think that table is better than figure.

Response 4: Thank you. We removed the figure and have inserted a new table (Table 2 is newly created in the revised version).

Point 5: In table 2, It seems that depression scale might not have a normal distribution. The authors should check normality of variables, and consider nonparametric test.

Response 5: Health-related quality of life and family concerns score were normally distributed, whereas depression score was right skewed. We have analysed again (Kruskal-Wallis test) and revised the Table 2 (Table 3 in the revised version) taking into account your critique.

Point 6: Because this study simply explored the association between traumatic event and work limitation using cross-sectional study, correction for multiple comparison (physical, psychosocial, and environmental limitation) might not be necessary in Table 3. However, the author should comment this issue in data analysis section.

Response 6: We analysed work limitations by dividing into three subdomains (physical, psychosocial, and environmental) in the Table 3 (Table 4 in the revised version, results of multivariable logistic regression), because the instrument “health-related work limitations questionnaire” was originally designed to be divided into three work demands and measured work limitations by type of work demand.

Point 7: English editing is required for this manuscript.

Response 7: We agree with the reviewer and the revised manuscript, the figure and tables are under English editing now.

Reviewer 2 Report

To:

Editorial Board

Title: “The Effect of the Exposure to Work-related Potentially Traumatic Events on Work Limitations among Firefighters: a Cross-sectional Study”

Dear Editor,

I read this manuscript and I think that:

-          The use of questionnaire can be considered as a limitation of the study design. This should be considered and discussed in a dedicated limitation section.

-          An example of the questionnaire should be included.

-          The English of the text should be revised due to typos.

-          Firefighters can be exposed to lung damages. How did the authors check for such occupational illness and more other related to their work?

-          What were the clinical characteristics of the study population? Did they take some drugs? Did they suffer from any kind of diseases? Please specify.

-          Inclusion and exclusion criteria should be better specified.

-          A flow chart of the study should be included.

-          The role of care manager should be discussed in such a context as it can be a useful professional figure for all of this kind of workers. Please discuss such a point in relation to the paper from Ciccone MM et al. Vasc Health Risk Manag. 2010 May 6;6:297-305.

Author Response

Response to Reviewer 2 Comments

Point 1: The use of questionnaire can be considered as a limitation of the study design. This should be considered and discussed in a dedicated limitation section.

Response 1: We agree with the reviewer and have added the following sentences in the discussion section (lines 306-314): “Second, the questionnaire in this study evaluates the validity of the work limitation—whether it is a work limitation because of the traumatic events. Although the survey was limited to trauma cases, other physical or mental conditions such as underlying diseases or occupational illness can also cause work limitations and the exact meaning can be diluted during the survey process. Third, the self-reported health-related work limitations questionnaire was the only instrument to assess firefighters’ work limitations in this study. It obscures the validity of the work limitations. Additional instruments such as checking firefighters’ absenteeism or presenteeism, conducting in-depth interviews, or using other work limitations questionnaires, which evaluates the work limitations more objectively are needed.”

Point 2: An example of the questionnaire should be included.

Response 2: We agree with the reviewer and have added some examples of the health-related work limitations questionnaire (lines 136-145).

Point 3: The English of the text should be revised due to typos.

Response 3: We agree with the reviewer and the revised manuscript, the figure and tables are under English editing now.

Point 4: Firefighters can be exposed to lung damages. How did the authors check for such occupational illness and more other related to their work?

Response 4: We agree with the reviewer. According to your critique, we have added the limitation we have in the discussion section (lines 306-310): “Second, the questionnaire in this study evaluates the validity of the work limitation—whether it is a work limitation because of the traumatic events. Although the survey was limited to trauma cases, other physical or mental conditions such as underlying diseases or occupational illness can also cause work limitations and the exact meaning can be diluted during the survey process.”

Point 5: What were the clinical characteristics of the study population? Did they take some drugs? Did they suffer from any kind of diseases? Please specify.

Response 5: We did not assess the clinical characteristics of the study participants. It could cause a bias, so we have describe the point in the discussion section (lines 306-310).

Point 6: Inclusion and exclusion criteria should be better specified.

Response 6: We agree with the reviewer and have rewritten the participants and procedure section and included a flowchart of study participation (lines 79-87, Figure 1).

Point 7: A flow chart of the study should be included.

Response 7: We agree with the reviewer and have included a flowchart of study participation (Figure 1).

Point 8: The role of care manager should be discussed in such a context as it can be a useful professional figure for all of this kind of workers. Please discuss such a point in relation to the paper from Ciccone MM et al. Vasc Health Risk Manag. 2010 May 6;6:297-305.

Response 8: Thank you. We have added it in the discussion section (lines 324-325).

Reviewer 3 Report

Line 42. Eliminate officers. Begin sentence with "In firefighting..."

Line 45 - why not use firefighters  throughout article instead of firefighting officers?

Line 52 - should write percentage instead of using symbol (%).

Line 52- should not begin a sentence with an acronym.

Line 55 and 56 - write percentage instead of symbol; do this throughout paper.

Lines 56 & 81 & 129 - single digits (7) should be written unless in a chart or table.

Line 65 for example is e.g., ...

Lin 97 needs amends.

Line 151 - Before using et al., all authors should be listed in a citation first.

Lines 190 to197 - Should introduce each table and figure and then provide a short summary after each table and figure.

Line 260 - U.S.

Line 278  * 284 - Use "because of" or the "result of" instead of "due to".  Due refers to dates and money due.

293 - When firefighters are exposed to job related traumatic events, there is a negative impact on their work.

Author Response

Response to Reviewer 3 Comments

Point 1: Line 42. Eliminate officers. Begin sentence with "In firefighting..."

Response 1: We agree with the reviewer and have changed the term “firefighting officers” to “firefighters” in the manuscript.

Point 2: Line 45 - why not use firefighters  throughout article instead of firefighting officers?

Response 2: We agree with the reviewer and have changed the term “firefighting officers” to “firefighters” in the manuscript.

Point 3: Line 52 - should write percentage instead of using symbol (%).

Response 3: We agree with the reviewer and have written “percent” or “percentage” instead of using symbol (%) in the manuscript.

Point 4: Line 52- should not begin a sentence with an acronym.

Response 4: We agree with the reviewer and have avoided beginning a sentence with an acronym in the manuscript.

Point 5: Line 55 and 56 - write percentage instead of symbol; do this throughout paper.

Response 5: We agree with the reviewer and have written “percent” or “percentage” instead of using symbol (%) in the manuscript.

Point 6: Lines 56 & 81 & 129 - single digits (7) should be written unless in a chart or table.

Response 6: We agree with the reviewer and have avoided using Arabic numerals for single digits in the manuscript.

Point 7: Line 65 for example is e.g., ...

Response 7: We agree with the reviewer and have corrected the expression (e.g.) in the manuscript.

Point 8: Lin 97 needs amends.

Response 8: We have corrected the line: “Work-related traumatic events” (line 104).

Point 9: Line 151 - Before using et al., all authors should be listed in a citation first.

Response 9: We have listed all authors at the first citation in the manuscript (line 123, 155, and 165).

Point 10: Lines 190 to197 - Should introduce each table and figure and then provide a short summary after each table and figure.

Response 10: We agree with the reviewer and have added the following sentences in the results section (lines 242-243): “Table 4 shows the risk of work limitations according to the type of traumatic events while on duty, based on the group with ‘none of the traumatic events’.”

Point 11: Line 260 - U.S.

Response 11: We have corrected it: “the U.S.” (line 289).

Point 12: Line 278  * 284 - Use "because of" or the "result of" instead of "due to".  Due refers to dates and money due.

Response 12: We agree with the reviewer and have corrected the conjunction (because of) in the manuscript (line 308, 317-318, and 323).

Point 13: 293 - When firefighters are exposed to job related traumatic events, there is a negative impact on their work.

Response 13: Thank you. The sentence needed correction. We have revised it (lines 327-328): “We demonstrated a relationship between firefighters’ exposure to work-related traumatic events and their work limitations.”

Round  2

Reviewer 1 Report

The authors revised their manuscript appropriately based on my previous comments. Thanks for giving me an opportunity to review this interesting paper and I have some minor comments. 

In manuscript, 'gender' should be changed to 'sex' (male and female) because the the authors simply used biological difference.

In Table 3, please, provide information on statistical method for p-value estimation in footnote. 

In Table 4, I think p-value might not be necessary. 

Author Response

Response to Reviewer 1 Comments

Point 1: In manuscript, 'gender' should be changed to 'sex' (male and female) because the the authors simply used biological difference.

Response 1: We agree with the reviewer and have changed the term “gender (men and women)” to “sex (male and female)” through the manuscript and tables.

Point 2: In Table 3, please, provide information on statistical method for p-value estimation in footnote.

Response 2: We agree with the reviewer and have added the information about statistical method for p-value estimation in footnote in the Table 2 and 3.

Point 3: In Table 4, I think p-value might not be necessary.

Response 3: We agree with the reviewer and have deleted p-value in the Table 4.

Reviewer 2 Report

To:

Editorial Board

Title: “The Association of the Exposure to Work-related Traumatic Events on Work Limitations among Firefighters: a Cross-sectional Study”

Dear Editor,

I read the revised version of this manuscript and I think that the authors well addressed my previous comments. The paper improved very much.

Author Response

Dear Reviewer

We appreciate you taking your time and reviewing our paper.

Thank you very much.